# An uncertainty-based model of the effects of fixation on choice

**Zhi-Wei Li**, **Wei Ji Ma** *

Center for Neural Science and Department of Psychology, New York University, New York, New York, United States of America

* weijima@nyu.edu

## Abstract

When people view a consumable item for a longer amount of time, they choose it more frequently; this also seems to be the direction of causality. The leading model of this effect is a drift-diffusion model with a fixation-based attentional bias. Here, we propose an explicitly Bayesian account for the same data. This account is based on the notion that the brain builds a posterior belief over the value of an item in the same way it would over a sensory variable. As the agent gathers evidence about the item from sensory observations and from retrieved memories, the posterior distribution narrows. We further postulate that the utility of an item is a weighted sum of the posterior mean and the negative posterior standard deviation, with the latter accounting for risk aversion. Fixating for longer can increase or decrease the posterior mean, but will inevitably lower the posterior standard deviation. This model fits the data better than the original attentional drift-diffusion model but worse than a variant with a collapsing bound. We discuss the often overlooked technical challenges in fitting models simultaneously to choice and response time data in the absence of an analytical expression. Our results hopefully contribute to emerging accounts of valuation as an inference process.

## Author summary

When people look longer at a food item, they tend to like it more. We propose a new theory in which this occurs because looking gathers information that reduces uncertainty, and people are uncertainty averse. We turn the theory into a mathematical model and fit it to previously published data. It fits better than the leading model, although we also find that the leading model can be improved.

## Introduction

Eye fixations affect choices in value-based decision-making. This was originally demonstrated in a consumer decision-making task by Krajbich and colleagues [1]. Their experiment consisted of two phases. In the rating phase, subjects rated 70 snack food items on a scale from -10 to 10. In the choice phase, subjects chose between two previously rated items. The authors found that subjects more often chose the item that they fixated on for longer, even when the

**Data Availability Statement:** The data and code code used for generating synthesized data, model fitting, and plotting is available on osf (DOI 10.17605/OSF.IO/8YAC9).

**Competing interests:** The authors have declared that no competing interests exist.

subjective ratings of the two items were equal. There is also evidence that the fixations directly cause the choice bias, instead of an underlying preference causing both [2, 3] (also see [4] for moral decisions and [5] for perceptual decisions; [6] for a review on causality between attention and choice).

To quantitatively describe the decision-making process, Krajbich and colleagues introduced the attentional drift-diffusion model (aDDM). This model predicts choices and reaction times based on the subjective ratings of the two items and the sequence of fixation times. Like the traditional DDM [7], the aDDM assumes a hypothetical decision variable that drifts noisily towards a bound corresponding to the item with higher subjective rating. Here, however, the drift is accelerated for the fixated item. A choice is made when the decision variable reaches one of two bounds (one for each item). The aDDM accounts for choice and response time data not only in binary choice [1], but also in ternary choice [8] and in purchasing choices [9].

The aDDM does not express choice behavior as the result of the maximization of a utility function. As such, it is somewhat disjoint from many other models in behavioral economics. To bridge this gap, we build on recent work in psychology that has formulated utility functions in value-based decision-making in terms of subjective beliefs [10–12]. In computer science, this notion is already much older, appearing for example in Bayesian Q-learning [13]. In these models, the agent maintains a continuum of hypotheses about value (or item attractiveness) and computes a Bayesian posterior distribution over value, which reflects the degree of belief in each value on the continuum; the studies differ in how the posterior is subsequently used. Intriguingly, dopamine neurons also seem to encode a distribution over future rewards [14].

These previous studies introduced the notion of probabilistic inference of value, but did not compare the resulting models to the aDDM in terms of their ability to fit data. Here, we propose a new value inference model that uses the posterior distribution as the basis of a utility function with an "uncertainty aversion" component. We fitted this model to the joint choice and response time data from Krajbich et al. (2010) [1], and formally compared our model against the aDDM. To anticipate our results: we cannot confidently conclude that our model fits better than aDDM, but we show that it is at least competitive, while arguably being more principled. We also point out overlooked technical intricacies in fitting both models, which might be of interest for future studies.

## Results

### Posterior-utility-choice (PUC) model

**Background: Posterior uncertainty about value.** We conceptualize the value of an item as the amount of future satisfaction resulting from consuming the item. An agent typically has incomplete or imperfect information about future satisfaction. For example, when choosing a food item, the saltiness, fat content, texture, etc. of the item are only approximately known, and so is the extent to which these properties will match the agent's personal preferences.

In the perception literature, incomplete and imperfect knowledge is most often modeled using the framework of Bayesian inference [15–18]. In this framework, the observer uses the sensory observations to build a subjective belief about a state of the world, as captured by a posterior probability distribution $p(\texttt{world state|observations})$. Recent work has proposed that value, despite being highly subjective and individualistic, may be inferred in much the same way as a world state in perception [10–12]; we use this notion as the basis of our model. In the Krajbich task, the agent can try to infer value from physical cues, such as product brand and information on the packaging, as well as from memories of previous experiences with the category [19]. Combining sensory and memory cues with knowledge of one's internal

**A**

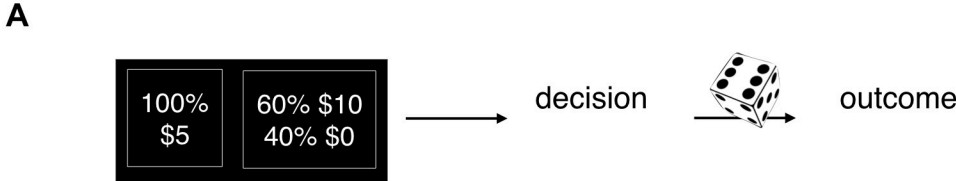

**B**

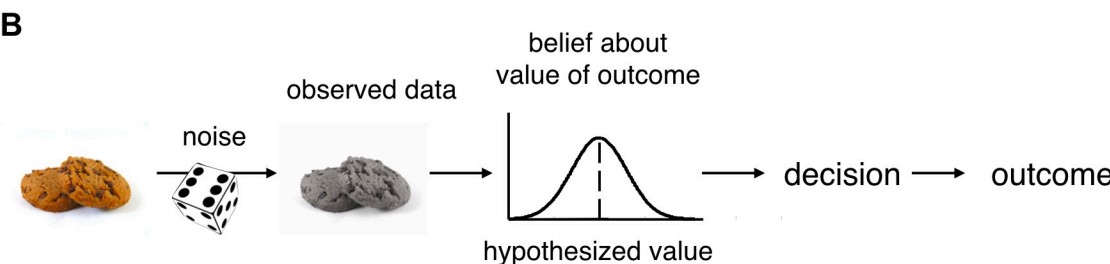

**Fig 1. Types of uncertainty in risky choice.** (A) In lotteries, uncertainty arises due to stochasticity in the mapping from decision to outcome. This is a form of aleatoric uncertainty. (B) Uncertainty can also arise from information about a choice item being incomplete or imperfect. In a probabilistic framework, the result of inferring the unknown value of a future outcome is captured by a posterior distribution, the width of which is a measure of uncertainty. This is a form of epistemic uncertainty.

state will allow the agent to form a belief about the value of the item. This belief can be expressed as a posterior probability distribution $p(\texttt{value}|\texttt{cues})$.

Based on this posterior distribution, posterior uncertainty can be defined as a summary statistic, for example as the standard deviation. Intuitively, when information is accumulated for a longer time, posterior uncertainty should generally decrease. Posterior uncertainty is different from classic risk, such as one when an agent chooses between two lotteries, e.g. $5 for sure, or a 50% chance of receiving $10 [20]. In such an experiment, the outcome is uncertain solely due to the stochastic step that follows the agent's choice. By contrast, posterior uncertainty can exist in a fully deterministic world. The distinction between posterior uncertainty and risk is an instance of the distinction between epistemic and aleatoric uncertainty, respectively [21]. We illustrate this distinction and how we view it in the full decision-making process in Fig 1.

**Model overview.** The basic premises of our model are as follows: a) fixating on an item will reduce posterior uncertainty about future satisfaction, either through the acquisition of visual information, or by triggering memory recall; b) lower uncertainty leads to higher utility in an uncertainty-averse agent. By combining these mechanisms, the model can in principle account for the effect of fixation on choice: longer fixation on an item leads to lower uncertainty, which leads to higher utility, which leads to a higher probability of choosing the item. We will now turn these premises into a concrete mathematical model, comment on the relations with other models, fit the model to Krajbich' data, and compare the fit with the fit of the aDDM.

We first specify the model mathematically. In the model, when viewing a food item, the agent is trying to predict the value (future satisfaction) derived from consuming the item. On a given trial, the decision process consists of three steps:

1. the computation of a **posterior** distribution over value (with associated posterior uncertainty);

2. converting a given posterior distribution over value into a scalar **utility**;

3. converting utility to **choice**.

Accordingly, we will refer to the model as the Posterior-Utility-Choice (PUC) model. The PUC model is partially normative because it computes a Bayesian posterior in Step 1 and a utility function in Step 2. However, the form of the utility function is ad-hoc, and unlike in recent work by Tajima et al. [10], Step 3 is not normative.

**Generative model.** Before we can describe the decision model, we need to specify the generative model, which describes the statistical origins of the agent's data. In PUC, step 1 (posterior computation) is directly based on the generative model.

We denote the state resulting from consuming a food item by $s$. This high-dimensional state could comprise many factors, including satiety, nutritional state, and health status. The agent, however, does not have direct access to $s$, but has to form a beliefs about $s$ from the available data $D$, which include observations of the properties of the item (size, ingredients, brand, etc.), interoceptive data about one's own homeostatic state [22], and memories of consuming similar items [19]. We assume that in a limited time, only a limited amount of imperfect data can be collected. The mapping from $s$ to $D$ is stochastic and can be described by a probability distribution $p(D|s)$.

Next, we assume that the state $s$ will produce—in a deterministic or stochastic fashion—a value $v$, representing an amount of future satisfaction. We model this mapping as a probability distribution $p(v|s)$. The graphical representation of the generative model is then $D \leftarrow s \rightarrow v$, where $s$ generates both $D$ and $v$.

**Decision model.** We now use the generative model to specify the agent's decision-making process. This process consists of three steps (P-U-C): from the sensory and memory data to a posterior distribution (Step 1), from a posterior distribution to utility (Step 2), and from utility to choice (Step 3). We now discuss each step.

- **Step 1: From data to posterior distribution over value**
  Given specific observed data $D_{\mathrm{obs}}$, the agent entertains a range of hypotheses about value. The likelihood of hypothesized value $v$, denoted by $L(v; D_{\mathrm{obs}})$, is now the probability that $D_{\mathrm{obs}}$ were produced by a state of value $v$:

$$L(v; D_{\mathrm{obs}}) = p(D_{\mathrm{obs}}|v) = \int p(D_{\mathrm{obs}}|s)p(s|v)ds.$$

  Here, $D_{\mathrm{obs}}$ and $s$ are both high-dimensional, whereas $v$ is one-dimensional. We assume that the effect of viewing longer is that more data are gathered. Unfortunately, we know neither $p(D_{\mathrm{obs}}|s)$ nor $p(s|v)$ since the researchers cannot observe the complete interoceptive state $s$. Therefore, we make a simplification, where we assume that the likelihood over $v$ is Gaussian with mean $x$ and standard deviation $\sigma$, which we will assume to be the same for all items:

$$L(v; D_{\mathrm{obs}}) \propto \mathcal{N}(v; x(D_{\mathrm{obs}}), \sigma(D_{\mathrm{obs}})^2).$$

  Here, $x$ is the maximum-likelihood estimate of $v$, the best guess one could make about $v$ based on $D_{\mathrm{obs}}$. By analogy with perception, we will refer to $x$ as a *measurement*, and a consistent generative model for $x$ would be $p(x|v) = \mathcal{N}(x; v, \sigma^2)$, where $v$ is the true value. This simplification is largely analogous to the mapping from a neural population model to a behavioral model [23]. As a proxy for the true value of consuming an item, we take the rating given by the subject for that item in the rating phase of the experiment.
  We next assume a Gaussian prior $p(v) = \mathcal{N}(v; \mu_p, \sigma_p^2)$. Including the prior, the posterior over $v$ becomes

$$p(v|D_{\mathrm{obs}}) \propto L(v; D_{\mathrm{obs}})p(v), \tag{1}$$

which we approximate by

$$p(v|x) \propto p(x|v)p(v). \tag{2}$$

The accumulation of evidence is gated by fixations. Moreover, the longer the agent fixates on an item, the more measurements are made. Within a trial, the number of measurements can differ between the two items due to unequal fixation times. For each item, the posterior distribution starts as the prior distribution. As the agent gathers measurements by fixating on an item, the posterior distribution for that item updates iteratively:

$$p(v|x_1, \ldots, x_{T+1}) \propto p(x_{T+1}|v)p(v|x_1, \ldots, x_T). \tag{3}$$

Qualitatively, this updating has two effects: first, the posterior mean $\mathbb{E}[v|x_1, \ldots, x_T]$ will move away from the mean of the prior, $\mu_p$, toward the true value, $v$; second, the variance of $v$ under the posterior distribution will decrease (Fig 2, step 1).

- **Step 2: From posterior distribution over value to utility**
  Now that the agent has a belief over value as expressed in the posterior distribution, they have to turn this belief into a utility. The simplest way is simply to take a central tendency of the posterior (mean, median, or mode of value under the posterior). But this might discard too much information. In principle, any mapping from the posterior to a number can serve to compute the utility of an item. Since the posterior is a function, utility is then a function of a function, and we will denote it by $U[p(v|\mathbf{x})]$.
  Furthermore, we define utility as a weighted average of the mean and the standard deviation of value under its posterior distribution based on a sequence of measurements $\mathbf{x} = (x_1, \ldots, x_T)$ (Fig 2B):

$$U[p(v|\mathbf{x})] = \mu_{\text{posterior}} - A \cdot \sigma_{\text{posterior}}, \tag{4}$$

where $A$ is a constant that we will call the "uncertainty aversion parameter", and

$$\mu_{\text{posterior}} = \mathbb{E}[v|\mathbf{x}] \tag{5}$$

$$\sigma_{\text{posterior}} = \text{SD}[v|\mathbf{x}] \tag{6}$$

The standard deviation term can be motivated in at least three, not mutually exclusive ways. First, it is possible that the proper definition of utility is $U = \mathbb{E}[v|D_{\text{obs}}]$, i.e. the expected value of the posterior based on $D_{\text{obs}}$. We made the approximation that the likelihood over $v$ is Gaussian, but if it is not, an error will be introduced by instead using $\mathbb{E}[v|x]$. The standard deviation term could partially compensate for this error, in a way similar to [24]. Second, even when the likelihood is exactly Gaussian, the agent might be uncertainty-averse. The uncertainty arises from incomplete and imperfect data (i.e. stochasticity in the measurements), rather than from post-decision stochasticity, as in classic risk aversion. Standard models of risky choice such as in Fig 1A are a specific case of our model. In the classic risk models, the future state $s$ is typically the state of having received a specific monetary amount, and no data need to be collected to know the distribution of $s$. Moreover, the mapping from $s$ to $v$ is typically assumed deterministic and monotonic, say $v = F(s)$. Then Eq (1) for the posterior over $v$ reduces to $p(v) = \int \delta(v - F(s))p(s)ds$ (all of this can be conditioned on an action). Thus, our model generalizes standard models of risky choice by replacing the distribution $p(v)$ by a posterior distribution $p(v|D_{\text{obs}})$. A term similar to the standard deviation term has been used in portfolio theory [25]. Third, we are inspired by the literature on the so-called *mere exposure effect* [26], the phenomenon that mere exposure to a stimulus (for example a

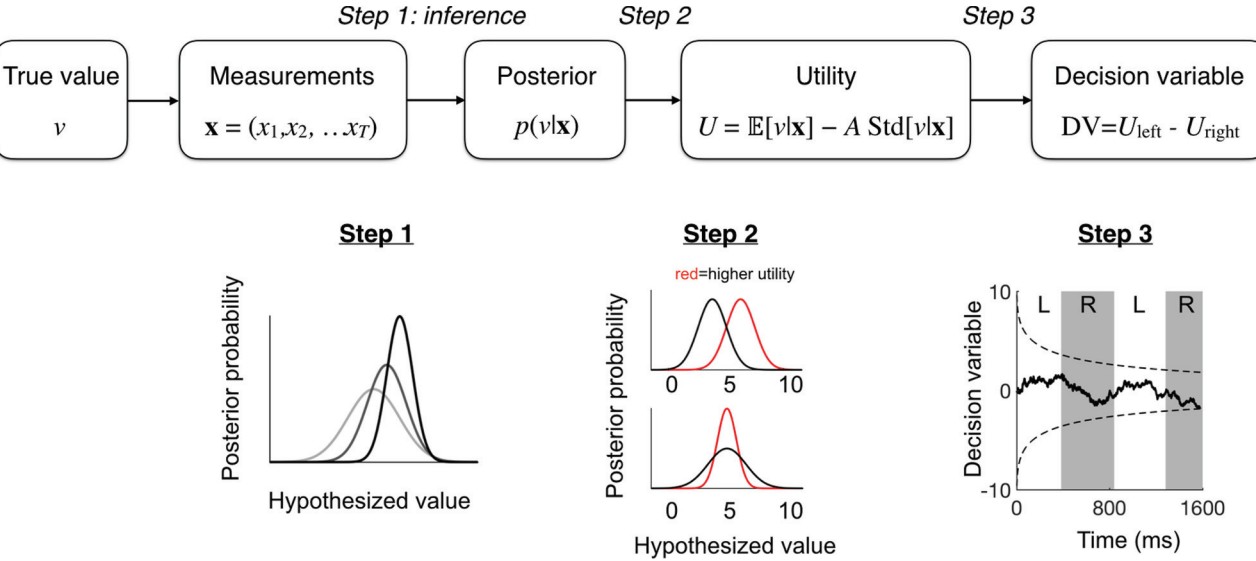

**Fig 2. Posterior-Utility-Choice model.** The PUC model describes how an agent maps noisy measurements of value to a decision variable. Top: Flow diagram of the model. Bottom: Components of the model. Step 1: The agent computes a posterior distribution over hypothesized value. As viewing time increases (darker colors), the posterior distribution shifts from the prior towards the true value, and becomes narrower. Step 2: Utility incorporates both the mean and the standard deviation of the posterior over value. Both higher mean and lower standard deviation are preferred. Utility is evaluated separately for each item. Step 3: Evolution of the decision variable on an example trial. L and R denote fixations on the left or the right item. The decision variable, DV, is the utility difference of the two items. A decision is made when DV crosses the collapsing bound (dashed).

consumer item or a work of art) increases the observer's preference for this stimulus. A leading explanation of the mere exposure effect is uncertainty reduction [27, 28], the idea that people prefer stimuli that are familiar.

Under our Gaussian assumptions for likelihood and prior, the mean and standard deviation of the posterior become

$$\mu_{\text{posterior}} = \frac{\frac{\mu_p}{\sigma_p^2} + \frac{T\bar{x}}{\sigma^2}}{\frac{1}{\sigma_p^2} + \frac{T}{\sigma^2}} \tag{7}$$

$$\sigma_{\text{posterior}} = \frac{1}{\sqrt{\frac{1}{\sigma_p^2} + \frac{T}{\sigma^2}}}, \tag{8}$$

where $\bar{x}$ is the mean of the measurements $(x_1, \ldots, x_T)$. Increasing viewing time will change utility in two ways: by moving the posterior mean from $\mu_p$ to the true value $v$, and by decreasing the posterior standard deviation. The former of these changes can be negative or positive depending on the prior estimation of value and the specific value of the current item, the latter is always positive. In Fig 2, step 2 we show two examples of how one item can have higher utility than the other.

- **Step 3: From utility to choice**
  The final step is to map utility to choice. We posit that the agent's decision variable, denoted by DV, is the difference between the utilities of the two items:

$$\text{DV} = U_{\text{left}} - U_{\text{right}}. \tag{9}$$

Finally, the agent terminates the decision process when DV crosses a decision bound. The

agent will then choose the item with the higher utility (Fig 2, step 3). We choose the decision bound to be decreasing over time ("collapsing"). A general motivation for using a collapsing bound instead of a fixed one is to prevent the model from predicting unrealistically long reaction times when deciding between two very similar items [29–32]. For the specific form of collapsing bound, we use a special case of the Weibull function suggested by [30]:

$$B_t = B_0 \, e^{-\left(\frac{t}{\lambda}\right)^k}. \tag{10}$$

An example of the evolution of the decision variable and the decision bound is shown in Fig 2, step 3).

In addition, we allow for non-decision (or residual) time $\tau$. This means that the decision process may end before the fixation series has ended, and the remaining time would then not contribute to the decision, regardless of how many fixations occur in that time; in practice, the estimated non-decision time is so short that fixation rarely switches in this time. Our non-decision time is conceptually different from the one in [8, 9]: theirs can be interpreted as accounting for the transitions between fixations, while ours is part of the total fixation time and interpreted as the time spent looking at items after the decision has been made. Finally, we add a guessing rate parameter $g$. For each trial, there is a probability $g$ of making a random decision at a random moment (based on the empirical decision time distribution fitted by a Weibull function, details see S2 Text), then the rest probability of $1 - l$ making decision based on the model prediction. This is necessary to avoid a zero probability for a smooth likelihood function landscape that allows parameter fitting.

## Attentional drift-diffusion model

Krajbich and colleagues [1, 8, 9, 33] have proposed the attentional drift diffusion model (aDDM), which conceptualizes the consequences of fixation as biasing the drift velocity of the fixated item. The decision variable is defined as:

$$DV_t = DV_{t-1} + d(r_{\text{left}} - \theta r_{\text{right}}) + \epsilon_t \tag{11}$$

when fixating on the left item, and

$$DV_t = DV_{t-1} + d(\theta r_{\text{left}} - r_{\text{right}}) + \epsilon_t \tag{12}$$

when fixating on the right item, where $d$ is a scaling constant, $r_{\text{left}}$ and $r_{\text{right}}$ are the ratings for the two items, and $\epsilon_t$ is diffusion noise, drawn independently across time points from a normal distribution $\mathcal{N}(0, \sigma^2)$. aDDM differs from the standard drift-diffusion model [7] in the attentional bias factor $\theta$, which takes values between 0 and 1. Diffusion continues until the DV hits one of two boundaries, which are assumed symmetric with respect to 0. In the original work by Krajbich and colleagues [1], the boundaries were fixed over time. As usual, there is an arbitrary scaling in aDDM which allows us to set $B_0 = 1$; in PUC, this is not possible because the scale is already set by $v$, which we approximate by the subject's rating of the item.

**aDDM with collapsing bounds.** We also consider a more flexible variant of the aDDM, namely one that has the same parametric family of collapsing bounds as the PUC model (Eq (10)). We call the resulting model the attentional collapsing-bound drift-diffusion model (acbDDM), by analogy to the non-attentional version, which has been called cbDDM, with "cb" standing for "collapsing bound". Milosavlevic et al. [31] previously considered a collapsing bound that is a special case of ours, namely with $k = 1$. In principle, we could also allow for an increasing bound by changing the parameter ranges of the boundary function. We limit

ourselves to a collapsing bound here, following previous work and because it is psychologically easier to interpret.

**Differences between PUC model and aDDM.** The PUC Model and the a(cb)DDM have mechanistic similarities: in both models, the agent decides when the decision variable crosses a bound, allowing the model to make predictions for the relation between choice and total fixation time. However, the PUC model differs conceptually from the a(cb)DDM in the following aspects: (a) the PUC agent chooses the item with the highest utility, whereas the a(cb)DDM does not have an immediate interpretation in terms of utility. (This stands in contrast to the basic DDM model, in which the decision variable can be interpreted as the difference between the values of the two items [34]); (b) in the PUC, noise is specifically associated with the agent's observed sensory information and retrieved memories, whereas the origin of noise in the a(cb)DDM is not well specified; (c) In the PUC model, later measurements have a smaller effect on the decision variable than earlier ones, because all measurements are generated from the same distribution and there are diminishing returns to information collection as the estimated value approaches the underlying true value. By contrast, in the a(cb)DDM, the variance of the noise added at each time point stays the same across time; (d) in the PUC model, two main mechanisms influence the preference: the uncertainty reduction term is independent of the item rating, indicating an additive effect of attention, whereas the posterior mean update depends on the difference between the prior mean value and the specific item value, thus indicating a multiplicative effect of attention. In contrast, a(cb)DDM will always boost the item with higher value more, thus the influence of attention is always multiplicative. For more studies regarding the additive versus multiplicative nature of attention, see [35–37].

In addition, on the surface, it seems that in the PUC model, the agent keeps two distinct value distributions, one for each item, whereas in the a(cb)DDM, the information of the two items are combined into a single "relative decision variable". However, in the later extension of aDDM model where more than two options are being compared ([8]), separate accumulators for each item were used and and it could be reduced to the aDDM with two items. Thus, the a(cb)DDM can also be conceptualized as two distinct accumulators, followed by another step of deriving the relative value difference, which is similar to the PUC model.

To model the effects of fixation on choice, we introduced the Posterior-Utility-Choice (PUC) model, in which the agent judges the value of an item by (1) accumulating evidence, gated by fixations; (2) computing and updating two posterior probability distributions over value, one for each item; (3) calculating the utility of each item not only from its posterior mean but also from its posterior standard deviation, with the latter accounting for uncertainty aversion. We compared the PUC model to the established attentional Drift Diffusion Model (aDDM). We fitted model parameters to individual-subject data using maximum-likelihood estimation.

**PUC versus aDDM.** To compare the goodness of fit of the PUC model with that of the aDDM, we first inspected model fits to several summary statistics plotted by [1] (Fig 3): the proportion of choices of an item as a function of the fixation time advantage for that item, the same but conditioned on the item's rating, and the proportion of choices of an item as a function of the rating difference between the two items and which item was fixated last. To obtain the fits of a model to a summary statistic, we ran each model in generative mode using the fitted parameters for an individual subject to obtain a model prediction for each trial, then aggregated these predictions to compute the summary statistic.

Both the PUC model and the aDDM qualitatively capture the behavioral phenomena that an item viewed for a longer time (Fig 3A and 3B) or viewed last (Fig 3C) is more likely to be chosen. However, both models show modest deviations from the data. In addition to choice

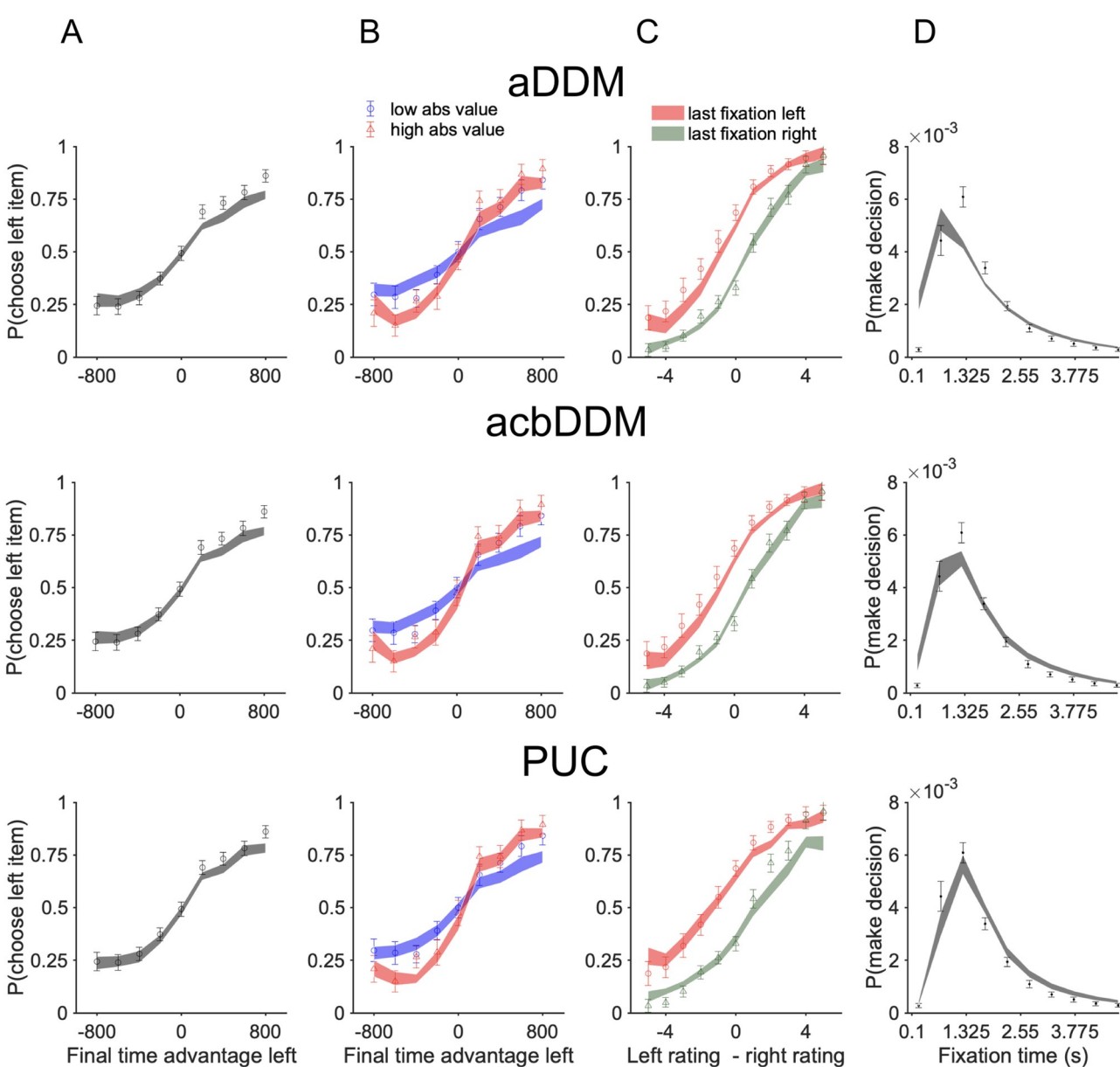

**Fig 3. Fits of the aDDM, the acbDDM, and the PUC model to summary statistics of the data.** (A) When the total fixation time advantage of an item increases, that item is chosen more often. (B) Same as A but conditioned on item rating. Both models predicted that when the absolute values of both items are higher (i.e. both are more preferred items), the fixation modulation effect is larger, which trend is less significant in the empirical data. (C) Besides total fixation duration, the last fixation also biases the choice. (D) Distribution of total fixation time. The aDDM fits we obtained differ from those in [1] not only because of differences in parameter estimation methods (see **Model fitting and model comparison—Differences from Krajbich et al. (2010)**), but also because of a difference in trial aggregation (see **Results—PUC versus aDDM**).

data, we also plotted the distribution of the total fixation time (Fig 3D), which shows a somewhat worse fit for the aDDM than for the PUC model.

Note that our fits of the aDDM to the summary statistics look different from those given by [1]. This is mainly because of a difference in trial aggregation. Krajbich et al. aggregated the model predictions across rating pairs without taking into account the frequencies of these pairs in the experiment. We instead aggregated individual-trial predictions; this is necessary in

**Table 1. Comparing the main PUC model to alternative models according to negative log likelihood (not corrected for the number of free parameters), AICc, and BIC.** Lower values are better for the first-mentioned model. All values are summed across subjects; bootstrapped 95% confidence intervals are given in parentheses.

|  | PUC-aDDM | PUC-acbDDM | acbDDM-aDDM |
| --- | --- | --- | --- |
| neg LL | -321 (-533, -160) | 127 (55, 197) | -448 (-613, -318) |
| AICc | -461 (-887, -136) | 253 (110, 394) | -714 (-1045, -456) |
| BIC | -288 (-708, 35) | 253 (110, 394) | -541 (-869, -284) |

our case because we used the individual-trial fixation series, but it also ensures a proportional representation of each rating pair in the summary statistics.

To quantitatively compare the models, we performed formal model comparison (Table 1). The PUC model has a lower AICc than the aDDM (thus a better fit), with a summed difference across subjects of −461. The bootstrapped 95% confidence interval was (−887, −136). BIC penalizes the number of parameters more, causing the 95% confidence interval to include 0. Overall, the PUC model performs better than the aDDM.

**Collapsing bound.** Next, we included the aDDM with collapsing bound, acbDDM, into the comparison. The fits of the acbDDM to the summary statistics are similar to those of the aDDM (Fig 3), but the acbDDM performs substantially better than the aDDM in model comparison (summed AICc difference: -714; summed BIC difference: -541). The acbDDM model also clearly outperforms the PUC model (summed AICc and BIC difference: 253; these two metrics are same because both models have the same number of parameters).

**Hierarchical Bayesian model selection.** So far, we assumed that every subject followed the same model. However, there might be heterogeneity in the population. To account for this possibility, we performed hierarchical Bayesian model selection [38] using the VBA package [39]. According to this analysis, the proportions of the population following the PUC, the aDDM, and the acbDDM are 20.5%, 25.6%, and 53.9% based on AICc (and 18.0%, 35.9%, and 46.1% based on BIC). However, more work would be needed to conclusively establish heterogeneity in the population.

**PUC model variants.** Finally, we examined three variants of the PUC model: first, a variant in which the prior variance is a free parameter instead of a fixed constant; second, a variant in which both the prior variance and the prior mean are free parameters; third, a variant without the uncertainty term in Eq (4) (i.e. $A = 0$). We found that all three variants fit worse than the main model we presented in the paper, in terms of both AICc and BIC (see S1 Text).

## Discussion

In this work, we use the recent idea that valuation is a form of Bayesian inference to explain the effect of fixation on choice. In the Posterior-Utility-Choice (PUC) model, the agent continuously updates a posterior distribution over the value of an item based on a sequence of noisy measurements and computes the utility of that item as a weighted difference between the posterior mean and posterior uncertainty, the latter reflecting uncertainty aversion. The decision is made when the utility difference between the two items reaches a bound. We found that the PUC model accounts better for the effects of fixation on choice than the original model, the attentional drift-diffusion model (aDDM), but not better than its generalization with a collapsing bound, the acbDDM; thus we provided some supports for a flexible bound as a decision model component (although, evidence is mixed in the literature [30, 31]).

Setting aside goodness of fit, the PUC model is a different type of model than the Attentional Drift-Diffusion models. The PUC model postulates what the agent cares about at a behavioral level, through a utility function derived from a posterior distribution over value. By

contrast, the a(cb)DDM is neither stated in terms of utility nor involves computing a belief over value. In addition, the PUC model makes a more explicit commitment to the origin of behavioral variability: ultimately, it stems from the noise in sensory measurements or retrieved memories. The a(cb)DDM does not make such a commitment.

These conceptual differences between the PUC model and the a(cb)DDM allow for novel predictions. First, the role of the uncertainty and therefore the utility of an item can be studied more explicitly in a new experimental design. Other than explicitly measure the uncertainty in the rating phase like in [40], future experiments could even manipulate the uncertainty by compromising the quality of the data that the subject receives about the value of an item. This could be done through a simple visual manipulation, such as lowering contrast or blurring the image, or through a memory manipulation, such as presenting items that differ in the time elapsed since the subject last interacted with them. We predict that lower quality of data will lead to lower utility and in turn to the item being less likely to be chosen. Second, a similar prediction could be tested in a timed rating experiment, where speeded judgments should lead to lower ratings. Third, we predict that changing the prior distribution in Eq (2) will affect choices in a specific manner. For example, consider a choice between two items with a true value of 3, yet the participant has a prior estimate of 0. Fixating longer on one item then has two effects on the utility of that item: it increases the posterior mean (since the likelihood mean is greater than the prior) and it decreases posterior uncertainty. Now consider the same agent but with a prior mean of 6. Then, longer fixation has two counteracting effects on utility: it will still decrease uncertainty, but now it will *decrease* the posterior mean. Comparing to the prior mean of 0, we expect a weaker or even reversed effect of fixation on choice. Thus, it might be interesting to experimentally manipulate the prior distribution.

Our work has several limitations:

- All extant models exhibit noticeable deviations from the data, which leaves a challenge for future modelers.

- We assumed a specific direction of causality: that fixation increases preference, instead of the other way round. Both the PUC model and the aDDM assume this causal direction, but the present data do not speak directly to this potential confound. Earlier work did to some extent: when the presentation times of items are controlled by the experimenter, the subjects will prefer the item with longer exposure duration [2]. However, it is not clear whether the magnitude of the effect is comparable between self-directed and passive fixation. This issue needs to be addressed experimentally.

- We assumed the prior to be Gaussian. Instead, one could allow for a richer parametrization of the prior or use an empirically grounded distribution as the prior. In addition, the agent might update the prior over the course of the experiment.

- Although Step 1 (computing the posterior) and Step 2 (computing the utility difference) of the PUC model are normative, Step 3 (the collapsing bound) is not. One could make this step normative for example by postulating that the agent maximizes expected reward rate [32], but this would make the model quite complicated without clear prospects for additional insight, in part because expected reward rate is only one way to take into account the cost of time.

- One could explore alternative forms of the uncertainty aversion term in Eq (4). We subtracted the standard deviation, but this is somewhat arbitrary. Instead, we could have subtracted a power of the standard deviation (e.g. variance), or added the inverse standard deviation.

- It is not clear how to generalize our model to the loss domain. For aversive items, the fixation bias seems to be in the opposite direction than for attractive items [3]; in other words, looking longer at an aversive item makes the item *less* likely to be chosen. An account for this effect could start from the finding that people tend to be risk-seeking in the loss domain [41]. Replacing risk attitude by uncertainty attitude, it is tempting to simply use $A < 0$ for aversive items in Eq (4) of the model. However, this would not produce a good process model, since it would be ill-specified how the agent sets the sign of $A$. Instead, we see greater promise in taking a step back and designing an alternative utility function, to replace Eq (4), that is the probability that the item under consideration has a value higher than a criterion $v_{\text{crit}}$:

$$U = \Pr(v > v_{\text{crit}} | \mathbf{x}). \tag{13}$$

If the posteriors are Gaussian, as in our model, this becomes

$$U = \Phi(\mu_{\text{posterior}}; v_{\text{crit}}, \sigma^2_{\text{posterior}}), \tag{14}$$

where $\Phi(\cdot; \cdot, \cdot)$ is the cumulative normal distribution with mean and variance parameters. The decision variable would still be the difference of the utilities of the left and right item, as in Eq (9). As an example, we now consider the case of $v_{\text{crit}} = 0$ and $\sigma_p \to \infty$, use the properties of the cumulative normal distribution, and substitute Eqs (7) and (8). This yields

$$U = \Phi\left(\frac{\mu_{\text{posterior}}}{\sigma_{\text{posterior}}}; 0, 1\right) = \Phi\left(\frac{\bar{x}\sqrt{T}}{\sigma}; 0, 1\right) \tag{15}$$

For positive $v$, this $U$ tends to increase with more observations, but for negative $v$, $U$ tends to decrease. Thus, an aversive item will become less preferred with longer looking time. We conclude that Eq (13) might provide a starting point for future models that generalize better to the loss domain (and that are also less arbitrary in the sense of the previous point).

Finally, we briefly address recent studies that also apply a value inference framework to understand attention-modulated decision-making [11, 42, 43]. These studies interpret the switching of attention as an active sampling process and derive the switching strategy from a optimal policy. The optimal policies are derived in different ways, some with an explicit decreasing threshold like in the PUC model [11], while others assume that the sampling and switching costs need to be balanced with accurate posterior estimation [42, 43]. In addition, these models differ in how the behavioral signature of "more fixated item being more preferred" is reproduced. One assumption that they shared is that the prior mean of the item value is either zero or lower than the true value ("prior bias" in [43]). As a result, the less sampled item will have the posterior value closer to prior mean rather than the true value which is higher than the prior. Jang et al. [42] also assumed that attention changes the precision of observations, so that the unattended item will incur samples with bigger variance, thus the expected mean will approach to the true mean even slower; this is similar to the PUC model. Song et al., on the other hand, assumed a distorted value perception for the unattended item by assuming a lower sample mean for that item [11].

A difference between these approaches and our model is that our reproduction of the qualitative effect does not require an assumption that the prior mean is zero or lower than the true mean; instead, we fix the prior mean to the empirical mean of the ratings on an individual basis. Instead, we introduced the subjective utility with an uncertainty aversion component, the underlying consideration being that consumer decisions may not merely amount to a value comparison similar to perceptual tasks, but also involve choice bias mechanisms. A

shortcoming of our approach, however, is that we are not able to predict fixation times. We believe it would be worthwhile to examine whether the PUC model can be equipped with an active sampling mechanism.

At a high level, our work fits in a broader set of recent attempts to appreciate the role of evidence accumulation and inference in value-based decision-making (other examples include [10, 12, 44]). We expect the probabilistic inference to become increasingly central in the study of valuation and decision-making.

## Materials and methods

### Data

The data from the experiment in [1] were made available to us by the authors. The data set contained 39 participants with an average of 95 trials per participant (25 participants completed the maximum number of 100 trials). On each trial, the data of interest consisted of the previously collected ratings of the presented items and the eye fixation series summarized as a binary sequence with values "left" and "right", with the corresponding fixation times. We are not fitting reaction times, but instead total fixation time.

### Likelihood

The PUC model, as introduced in "Decision models" above, has 4 parameters for the value estimation ($\sigma$, $\sigma_p$, $\mu_p$, $A$), three bound parameters ($B_0$, $k$, and $\lambda$), one guessing rate parameter ($g$) and a non-decision time parameter $\tau$. To simplify, we fixed the prior mean $\sigma_p$ and variance $\mu_p$ to be the empirical mean and variance extracted from the rating data, thus leaving 5 parameters to be fitted. We tested the more flexible versions of PUC too, as well as another reduced PUC model(see summary in the "Result" section and details in S1 Text). The aDDM has 3 parameters for the drift process: $\sigma$, $d$, $\theta$, two bound parameters ($k$ and $\lambda$, since in their model, $B_0$ can be set to 1 without loss of generality). To match with PUC for a fair model comparison, we added a guessing rate parameter ($g$) and a non-decision time parameter $\tau$ for aDDM.

We fitted the parameters in each model on an individual-subject basis using maximum-likelihood estimation. The inputs to the model on the $i$th trial consist of the ratings of the left and right items, $r_{\text{left},i}$ and $r_{\text{right},i}$. The model predicts the joint probability of the choice $C_i$ and the total fixation time $T_i$. The log likelihood of the parameters is then

$$\log L(\text{parameters}) = \sum_{i=1}^{n_{\text{trials}}} \log p(C_i, T_i | r_{\text{left},i}, r_{\text{left},i}, \text{fixation series}_i, \text{parameters}). \tag{16}$$

We did not constrain $T_i$ to be later than all of the observed fixations. Thus, the model is allowed to wrongly predict a total fixation time that falls somewhere in the middle of the fixation series.

### Fitting procedure

We used maximum-likelihood estimation to fit the parameters, separately for each participant. This involves maximizing Eq (16), which in turn involves calculating for a given parameter combination and on each trial the probability that the model observer produces the participant's choice $C_i$ with the participants' total fixation time $T_i$. To calculate this probability, we numerically propagated the probability distribution of the decision variable (See Eqs 9, 11 and 12) across time. At each time step, we used the boundaries to truncate the distribution, with the truncated probability being our estimate of the probability of the corresponding response. The (non-normalized) remaining part of the distribution is propagated further.

To maximize the log likelihood, we used Bayesian Adaptive Direct Search [45], a global optimization algorithm that uses Bayesian methods to approximate the shape of the likelihood function. To minimize the risk of getting stuck in local optima, we ran the optimization algorithm with multiple initial conditions (see S2 Text).

**Differences from Krajbich et al. (2010).** Our fitting procedure differs from the one used in the original paper [1] in the following aspects:

- Instead of fitting parameters at the group level, we fitted parameters to individual subjects. This is more accurate if individuals differ from each other.

- Krajbich et al. binned the data in 100 ms bins, but simulated the time course of their model evolution in 1 ms steps. Instead, we used steps of 100 ms for the latter as well. Apart from saving computational time, this choice is more consistent with the assumption that measurements are independent across time points (in view of neural autocorrelation functions). Moreover, 100 ms is still reasonably fine compared to the median total fixation time (which was about 1.4 seconds; mean being 1.9 seconds).

- Instead of fitting the reaction time, we fitted total fixation time, in an attempt to avoid epochs in which the observer did not fixate on either item.

- Krajbich et al. used randomly sampled fixation durations from the empirical distribution to perform the simulation both in this and later work [8, 9]. Instead, we used the actual fixation data for each trial, simulating only until the time when the actual fixation has ended and calculating the probability that the choice was made at the end of the empirical fixation series. All remaining probability went into a single bin representing later decision times. Note one exception is when we plot the summary statistics of probability of making choice against the total fixation time. We used the distribution of fixation times split out by subject, then independently and sequentially drew from this distribution to create a synthetic fixation series for each trial after the empirical fixation series has ended; we repeated this 10 times for each trial which is enough to achieve a stable result. This allows us to obtain an unrestricted model prediction for total fixation times (which we use as a proxy for reaction times throughout the paper).

- Krajbich et al. performed 1000 simulations to calculate the log likelihood for each rating pair (3000 in their later work [9]). However, this number is low relative to the number of possible responses (in Krajbich et al., 2 choices times 52 reaction time bins; for us, 2 choices times 50 total fixation time quantiles). In such situations, estimating the log likelihood through brute-force simulation not only causes variance to be high, but is also biased due to the nonlinearity of the logarithm [46]. This problem is particularly stark when the simulation assigns zero samples to an observed response. Therefore, instead, we used the numerical approximation mentioned above.

- Instead of using a grid search in parameter space, we used Bayesian Adaptive Direct Search [45], which is a more precise and more reliable optimization method.

**Parameter recovery and model checking.** To confirm the validity of our model fitting choices, we fitted synthetic data using the same fitting procedures as for the real data. To generate synthetic data, we used the exact same rating distribution as the real data by matching each synthetic trial with a real trial. Each synthetic subject was given the parameters that best fitted one real subject. Then we performed fitting for individual synthetic subjects using methods introduced above. Results are presented in S3 Text. The summary statistics are recovered

very well. Parameter recovery is generally good but somewhat worse for the more complex models (acbDDM and PUC). This is likely due to soft trade-offs between parameters. As a result, the parameter estimates in the real data should be taken with a grain of salt. However, the results of our paper do not rely on parameter estimates but only on log likelihoods and summary statistics. Therefore, the results are not affected by issues with parameter recovery.

**Model comparison.** To compare models, we used the corrected Akaike Information Criterion (AICc; [47, 48]) and the Bayesian Information Criterion (BIC; [49]).

## Supporting information

**S1 Text. PUC model variants.**
(PDF)

**S2 Text. Details of model fitting.**
(PDF)

**S3 Text. Parameter recovery.**
(PDF)

## Author Contributions

**Conceptualization:** Zhi-Wei Li, Wei Ji Ma.

**Formal analysis:** Zhi-Wei Li, Wei Ji Ma.

**Methodology:** Zhi-Wei Li, Wei Ji Ma.

**Supervision:** Wei Ji Ma.

**Visualization:** Zhi-Wei Li, Wei Ji Ma.

**Writing – original draft:** Zhi-Wei Li, Wei Ji Ma.

**Writing – review & editing:** Zhi-Wei Li, Wei Ji Ma.

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
