## [Decision Letter · Decision Letter 0]

9 Oct 2020

Dear Dr. Ma,

Thank you very much for submitting your manuscript "A posterior-based model of the effects of fixation on choice" for consideration at PLOS Computational Biology.

As with all papers reviewed by the journal, your manuscript was reviewed by members of the editorial board and by several independent reviewers. In light of the reviews (below this email), we would like to invite the resubmission of a significantly-revised version that takes into account the reviewers' comments. I thought the comments are reasonable and clear, so I don't need to expand on them.

My only meta-comment concerns R2's first point: it's not clear to me that a Gaussian rating distribution is necessary to validate the model. If a subject's internal model and the true distribution over values are both Gaussian, then it is true you would expect a Gaussian rating distribution, but it's possible for the Gaussian assumptions to be a good description of the subject's internal model even if the true value distribution is non-Gaussian, in which case you would expect a non-Gaussian rating distribution. Another way of saying this is that Bayesian models of cognition need not be well-calibrated. It's certainly nice when they are, but finding poor calibration is not by itself evidence against a Bayesian model. It's especially tricky in this context even just to measure calibration because we don't have independent access to the true values. This means that finding a non-Gaussian rating distribution could* *mean the prior is non-Gaussian, or the observation noise is non-Gaussian, or the true distribution over values is non-Gaussian, or any combination of these. In any case, it's worth checking --- you might get lucky!

We cannot make any decision about publication until we have seen the revised manuscript and your response to the reviewers' comments. Your revised manuscript is also likely to be sent to reviewers for further evaluation.

Sincerely,

Samuel J. Gershman

Deputy Editor

PLOS Computational Biology

Reviewer's Responses to Questions

**Comments to the Authors:**

Reviewer #1: This is a really nice paper investigating a Bayesian approach to value-based decision-making. Drift-diffusion models (DDM) assume that evidence is accumulated over time for each alternative. The attentional DDM (aDDM) assumes that the rate of evidence accumulation increases for the alternative that is attended to. This leads to a choice bias in favor of attended items. This paper proposes an interesting alternative model for this phenomenon. The model assumes that subjects have a prior expected value (and associated variance) for each alternative. When the subject attends to an alternative, they update its expected value and reduce the variance around that expectation. Subjects are risk-averse and so they prefer alternatives that they are more certain about. The preference for higher values and higher certainty leads subjects to choose alternatives that are higher valued but also attended to longer.

The model presented in this paper is very sensible and has a lot of nice features. It blends economic intuition about decision-making under uncertainty with cognitive knowledge about attention and information gathering. While the model does not substantially outperform the existing model (aDDM) in goodness-of-fit, it is still an important step, for at least a couple of reasons. First, the aDDM is silent about how/why decision-makers sample information. The authors’ PUC model makes it clear that the point of attending to an alternative is to get a more precise and accurate estimate of the value of that alternative. This intuition is critical to advance our understanding of how people sample information. While this paper does not attempt to model the fixation process, one can see how you would use this framework to tackle that question. Second, the authors’ model seems to (marginally) outperform the standard aDDM. Only when the authors give the aDDM collapsing bounds do the models perform similarly well. Thus, another takeaway point of the paper is that collapsing bounds are an important feature to consider in these model-comparison exercises, and they may be an important addition to the aDDM (though visually the improvements were imperceptible, see Fig. 3).

In summary, I think this is an important and well written paper. I have a number of mostly minor comments. If the authors can address those comments, I would be happy to endorse publication.

(1) Regarding the comparison between the aDDM and cbaDDM: Figure 3 shows little difference between the models. These are all choice plots, where collapsing bounds will have little impact. It might be useful to add a fourth column comparing the reaction time curves with each model compared to the data.

(2) In comparing the advantages of the PUC to the aDDM, the authors repeatedly state that in the aDDM the “meaning of the decision variable is unclear”. I’m not really sure what the authors mean by this. In the (a)DDM, the goal is to choose the better item. That is why drift rate is driven primarily by the alternatives’ values; this leads to the decision-maker choosing the higher-value item in most cases. As outlined by Webb (2019) in Management Science, the DDM is essentially just a choice rule that takes utilities as inputs and outputs logit choices. In other words, the authors’ model first updates values then calculates utilities, while the DDM first generates utilities then updates the decision variable. It isn’t that one model is more normative than the other, its that the models do things in a different order.

(3) More generally, I think that the authors claims about what is normative (or what isn’t) should be set aside. What is normative depends on lots of assumptions. The aDDM could certainly be normative if attention switching is costly and attention is limited; we know that the standard DDM is normative under certain conditions (Wald,1945). The authors’ model is already admittedly semi-normative, and that doesn’t even consider the fact that risk-aversion is not normative per se (though it is standard) and that there is no explanation in this model for the fixation series themselves, which seemingly lead to non-normative information gathering.

I would instead focus on the fact that this model is explicitly Bayesian and so is set up to tackle important questions of how attention should be allocated, how to incorporate prior information about the alternatives, what confidence ratings the model should produce, etc.

(4) The authors mention at some point that, unlike in the aDDM, later evidence has a smaller effect on the decision variable. Could the authors elaborate on this point? In Bayesian models the order of data is not supposed to matter. Is that violated here? Or do the authors simply mean that there are diminishing returns to information collection from the same alternative?

(5) In simulating their model, the authors use 100ms time steps, which is much larger than other papers in the literature. Did the authors verify that such large time steps do not lead to distortions in the simulated data? In particular, did the authors do any parameter recovery exercises to ensure that this method was effective?

(6) On a related note, in their simulations the authors use the empirical fixation series. What happened when the observed fixation series terminated (in the data) and a decision hadn’t yet been reached in the simulation? Did the decision suddenly end or were additional fixations generated in some way?

(7) I found the simplified PUCp to be a strange model to consider. Why would subjects have a prior of 0 when only appetitive items were presented in the experiment? Even if subjects for some reason started the experiment with this prior, one would think it would change after the rating task, or at least after a handful of choice trials. Why not instead try a model where the prior mean is the mean of each subject’s rating distribution? With a prior of 0, you are essentially guaranteeing that more attention will lead to a higher choice probability. The whole thing seems quite anti-Bayesian. To be clear, I’m not demanding that the authors test such a model, I would just like more explanation why they are even considering this prior=0 model. What would be worth doing is looking at the relation between the recovered mu_p parameters and subjects’ actual mean ratings. On that note, it would be useful to see the distributions of recovered parameters for each of the models (in the appendix would be fine).

(8) There has been a debate about the multiplicative vs. additive nature of attention in these choice processes, with Cavanagh et al. 2014 (JEP: General), Smith & Krajbich 2019 (Psych Sci), and Westbrook et al. 2020 (Science). I wonder if the authors could comment on how their findings might inform this debate. Their model would seem to fall somewhere in between, with the uncertainty reduction being “additive” (independent of value), while the benefit of updating from the prior would be “multiplicative”, in the sense that it is more beneficial the farther the true value is from the prior.

Minor comments

(9) There are some references the authors might find useful (in addition to the ones mentioned above).

Gwinn & Krajbich 2020 (JESP) - they discuss how certainty (as well as accessibility) about item ratings affects choice outcomes.

Parnamets et al. 2015 (PNAS) - they provide evidence for how exogenous manipulations of attention affect choice outcomes

Tavares et al. 2017 (Frontiers) - they essentially replicate Parnamets, but with perceptual stimuli

There are also a number of papers on these attention effects that the authors might want to consider including in their references. See Krajbich 2019 (Current Opinion in Psychology) for a review.

(10) I’m not sure, but I’m concerned there may be an error in Equation 7. It would seem that as T goes to infinity (as deliberation time increases), mu_posterior goes to zero. But perhaps I am missing something.

(11) Non-decision time is also thought to account for early orienting to the stimuli, in addition to motor latency at the end of the decision.

(12) In the aDDM, d is a scaling constant, not the drift rate per se. The drift is d multiplied by the attention-weighted values.

(13) The authors say that in the data the median RT was 4 seconds. Can they elaborate on this, given that in Krajbich et al. 2010, mean RT appears to be about 2 seconds?

(14) The authors claim that Krajbich et al. 2010 used a “fixation model” in their simulations, but in the paper they state that they “sampl[ed] fixation lengths from the empirical fixation data….”

(15) I think it would be more accurate to state that there is “some” evidence for collapsing bounds. Milosvaljevic et al. 2010 argue that collapsing bounds do not help once across trial drift variability is accounted for. Hawkins et al. 2015 show mixed evidence for collapsing bounds. And here, while collapsing bounds do seem to help in the goodness-of-fits statistics, qualitatively the improvement is imperceptible, at least in the plots the authors have shown.

(16) Prospect theory states that risk-seeking occurs in the loss domain because of diminishing marginal sensitivity, not “so that they will still have some probability to obtain a good outcome.” So while this is an interesting way to think about the loss domain, it is not based on the standard interpretation.

(17) Table 2 - please briefly describe the models in the table caption.

Reviewer #2: In this work, Li and Ma propose a partial normative model to study the effects of fixation on decision-making. In the model they propose, the utility is computed based on weighted sum of the posterior mean and standard deviation of the posterior distribution. These values depend on both the alternative input values and the amount of time fixating at each alternative. I think that this is an interesting attempt to formalize the originally proposed instantiation of the aDDM. There are however a couple of critical aspects that the authors may want to clarify related key assumptions adopted in their models and parameter fitting.

Major comments

1) The authors adopt a Bayesian approach to estimate a posterior distribution, which ultimately leads to the computation of an utility that is then used to compute the decision variable (by comparing the utilities of each alternative). The authors assume a Gaussian prior and a Gaussian likelihood function with constant noise (irrespective of value input). The authors use the choice data to fit both the prior (mean and variance), and the variance in the likelihood function. Here there are two critical issues. First, given that the authors fit both prior and likelihood using the choice data, there must be at least some evidence that the use of this prior shape (and the resulting fit) is valid. Do the rating data follow a Gaussian distribution? Is the estimated prior distribution comparable to the rating data distribution? It is essential that the authors show and report this information in the article. I appreciate that the authors briefly mention in the discussion section the fact that the Gaussian distribution assumption is a limitation. However, in my opinion, such strong assumptions should be backed up by a principled argument otherwise the Bayesian model ceases to be “normative”. Also, the motivation for such assumption should be stated already in the description of the model. This is not to say that I am against Bayesian approaches (actually quite the opposite), but this is precisely the issue that one sector cognitive psychology has used as ammunition against Bayesian approaches in our field. The authors could for instance obtain the best possible fit of a Gaussian distribution to the rating data and then use this as prior distribution to fit the choice data. Alternatively, out of sample fits could be adopted to estimate out of sample priors (however, this might not sully solve the problem).

2) The model proposed by the authors has 8 free parameters that are used to fit about 95 choice trials per participant. It has been previously argued that the reliability of fitting that many parameters to DDMs (in particular with dynamic bounds) can be problematic and might be not identifiable, in particular for low number of trials as is the case here. A more critical problem is that the authors are also freely fitting the prior distribution. As it has been shown in previous work, some DDM parameters and in particular boundary parameters are very much related to prior distributions (see related work by Drugowitsch). This in addition to the prior fitting issue highlighted in my previous point 1. Here it is essential to demonstrate via simulations that the Bayesian inference parameters and the boundary parameters are identifiable using the same number of trials used to fit the real data (i.e., about 100 trials). Without this evidence, it is difficult to evaluate whether all free model parameters assumed by the authors are jointly identifiable.

Minor comment

3) It would be informative to study further qualitative predictions that could uniquely emerge in the model proposed by the authors. At present it is not clear (at least qualitatively) what the key differences are between the PUC and the aDDM. For instance, it is possible that the model makes different predictions for long versus short trials as in the PUC the model is directly influenced by viewing time?

**Have all data underlying the figures and results presented in the manuscript been provided?**

Reviewer #1: **No: **Data are available from the original authors.

Reviewer #2: Yes

PLOS authors have the option to publish the peer review history of their article (what does this mean?). If published, this will include your full peer review and any attached files.

Reviewer #1: No

Reviewer #2: No
---

## [Decision Letter · Decision Letter 1]

14 May 2021

Dear Dr. Ma,

Thank you very much for submitting your manuscript "An uncertainty-based model of the effects of fixation on choice" for consideration at PLOS Computational Biology. As with all papers reviewed by the journal, your manuscript was reviewed by members of the editorial board and by several independent reviewers. The reviewers appreciated the attention to an important topic. Based on the reviews, we are likely to accept this manuscript for publication, providing that you modify the manuscript according to the review recommendations.

Sincerely,

Samuel J. Gershman

Deputy Editor

PLOS Computational Biology

[LINK]

Reviewer's Responses to Questions

**Comments to the Authors:**

Reviewer #1: The authors have done a quite thorough job addressing my comments from the previous version. There are just a few minor points that I think should be corrected before publication.

(1) One of the takeaway points of the paper is that collapsing bounds are important, as they improve the aDDM fits substantially. However, one issue with this claim is that the boundary parameters k and lambda were constrained to be positive in the fits, and there appears to be bunching of the estimates close to or at zero. This constraint essentially guarantees that you will find support for collapsing bounds as long as there is variability across participants. I'm not asking the authors to refit all the models, but if they don't, I think they need to qualify that they did not allow for increasing bounds over time. What they have shown is that allowing bounds to vary improves model fit, but not that bounds are necessarily collapsing.

(2) I think the authors need to be clearer about the parameter recovery results in the main text. I wouldn't say that parameter recovery is "somewhat worse" for the more complex models; it is non-existent. To my eye the correlation between true and fitted parameters looks to be about 0 for several of the parameters, for example lambda and k in the PUC model. While this might not impact model comparison, it does mean that these parameters can't really be interpreted, so readers should not try to fit these models to data and compare parameter values (as readers are wont to do).

Side note - where do the "original" parameter values come from anyways? They don't appear to be from the participants' best fitting values.

(3) In the "Differences between PUC and aDDM" section, point (c) says that in the aDDM the information about the two items is combined into a single variable while in the PUC there are two distinct value distributions. However, Krajbich & Rangel 2011 showed that the multi-alternative version of aDDM has separate accumulators for each item and that it reduces to the aDDM with two items. In other words, the aDDM can be thought of as two distinct accumulators with a downstream difference calculation to compute the relative variable. In other words, both models allow the agent to keep two distinct value representations, and in the multi-alternative case this is required. I would suggest removing point (c).

Minor comments

What are the units on d? I guess this is in "evidence per 100ms"?

aDDM B set to 0? I think you mean 1?

The PUC has 9 parameters, you fix 2, but that leaves 7 not 5…

The "DDM2 parameter recovery" figure has one panel that says “lps” with unreadable units - should that be k?

The PUC parameter recovery figure has a panel labeled “ObsVar” - should that be sigma?

Across the parameter recovery and parameter distribution figures, you are inconsistent in whether you use the greek symbol itself or spell it out.

Reviewer #2: I commend authors for their efforts to carefully address my concerns and questions.

The authors did an excellent job responding to my concerns, and I do not have any further comments.

**Have the authors made all data and (if applicable) computational code underlying the findings in their manuscript fully available?**

Reviewer #1: Yes

Reviewer #2: Yes

PLOS authors have the option to publish the peer review history of their article (what does this mean?). If published, this will include your full peer review and any attached files.

Reviewer #1: No

Reviewer #2: **Yes: **Rafael Polania

Figure Files:

Data Requirements:

Reproducibility:

References:

---

## [Editor Report · Decision Letter 2]

17 Jun 2021

Dear Dr. Ma,

We are pleased to inform you that your manuscript 'An uncertainty-based model of the effects of fixation on choice' has been provisionally accepted for publication in PLOS Computational Biology.

Best regards,

Samuel J. Gershman

Deputy Editor

PLOS Computational Biology

---

## [Editor Report · Acceptance letter]

11 Aug 2021

PCOMPBIOL-D-20-01716R2 

An uncertainty-based model of the effects of fixation on choice

Dear Dr Ma,

I am pleased to inform you that your manuscript has been formally accepted for publication in PLOS Computational Biology. Your manuscript is now with our production department and you will be notified of the publication date in due course.

With kind regards,

Andrea Szabo
